# In Vivo PET Imaging of Monocytes Labeled with [^89^Zr]Zr-PLGA-NH_2_ Nanoparticles in Tumor and *Staphylococcus aureus* Infection Models

**DOI:** 10.3390/cancers13205069

**Published:** 2021-10-10

**Authors:** Massis Krekorian, Kimberley R. G. Cortenbach, Milou Boswinkel, Annemarie Kip, Gerben M. Franssen, Andor Veltien, Tom W. J. Scheenen, René Raavé, Nicolaas Koen van Riessen, Mangala Srinivas, Ingrid Jolanda M. de Vries, Carl G. Figdor, Erik H. J. G. Aarntzen, Sandra Heskamp

**Affiliations:** 1Department of Tumor Immunology, Radboud Institute for Molecular Life Sciences, Radboud University Medical Center, Geert Grooteplein 28, 6525 GA Nijmegen, The Netherlands; kim.cortenbach@radboudumc.nl (K.R.G.C.); Koen.vanRiessen@radboudumc.nl (N.K.v.R.); mangala.srinivas@wur.nl (M.S.); Jolanda.deVries@radboudumc.nl (I.J.M.d.V.); Carl.Figdor@radboudumc.nl (C.G.F.); 2Department of Medical Imaging, Radboud University Medical Center, Geert Grooteplein Zuid 10, 6525 GA Nijmegen, The Netherlands; Milou.Boswinkel@radboudumc.nl (M.B.); akip@citryll.com (A.K.); Gerben.Franssen@radboudumc.nl (G.M.F.); Andor.Veltien@radboudumc.nl (A.V.); Tom.Scheenen@radboudumc.nl (T.W.J.S.); Rene.Raave@radboudumc.nl (R.R.); Erik.Aarntzen@radboudumc.nl (E.H.J.G.A.); Sandra.Heskamp@radboudumc.nl (S.H.); 3Cenya Imaging BV, Tweede Kostverlorenkade 11H, 1052 RK Amsterdam, The Netherlands

**Keywords:** PLGA-NH_2_ nanoparticles, PET/MRI imaging, zirconium-89 labeling, in vivo tracking, primary amine, cell labeling

## Abstract

**Simple Summary:**

Immune cells are increasingly used for therapy in cancer and other diseases. To better understand immune-cell kinetics, cell-tracking with highly sensitive imaging modalities is required. The aim of this study was to develop a new strategy for the in vivo tracking of a small number of cells, using positron emission tomography (PET). We labeled poly(lactic-*co*-glycolic acid) nanoparticles containing a primary endcap (PLGA-NH_2_) with the radionuclide zirconium-89. The nanoparticles were characterized for size, polydispersity index, zetapotential and radiolabel retention. Subsequently, they were used for the ex vivo radiolabeling of a monocyte cell line (THP-1). We demonstrated that these radiolabeled monocyte cells can be traced in vivo in mouse tumor and infection models.

**Abstract:**

The exponential growth of research on cell-based therapy is in major need of reliable and sensitive tracking of a small number of therapeutic cells to improve our understanding of the in vivo cell-targeting properties. ^111^In-labeled poly(lactic-*co*-glycolic acid) with a primary amine endcap nanoparticles ([^111^In]In-PLGA-NH_2_ NPs) were previously used for cell labeling and in vivo tracking, using SPECT/CT imaging. However, to detect a low number of cells, a higher sensitivity of PET is preferred. Therefore, we developed ^89^Zr-labeled NPs for ex vivo cell labeling and in vivo cell tracking, using PET/MRI. We intrinsically and efficiently labeled PLGA-NH_2_ NPs with [^89^Zr]ZrCl_4_. In vitro, [^89^Zr]Zr-PLGA-NH_2_ NPs retained the radionuclide over a period of 2 weeks in PBS and human serum. THP-1 (human monocyte cell line) cells could be labeled with the NPs and retained the radionuclide over a period of 2 days, with no negative effect on cell viability (specific activity 279 ± 10 kBq/10^6^ cells). PET/MRI imaging could detect low numbers of [^89^Zr]Zr-THP-1 cells (10,000 and 100,000 cells) injected subcutaneously in Matrigel. Last, in vivo tracking of the [^89^Zr]Zr-THP-1 cells upon intravenous injection showed specific accumulation in local intramuscular *Staphylococcus aureus* infection and infiltration into MDA-MB-231 tumors. In conclusion, we showed that [^89^Zr]Zr-PLGA-NH_2_ NPs can be used for immune-cell labeling and subsequent in vivo tracking of a small number of cells in different disease models.

## 1. Introduction

Cell-based therapy is maturing into clinical practice and holds great promise for treating cancer, as well as immune-related diseases. In vivo cell tracking is desired to better understand the complex cell targeting mechanism and cell–cell interactions. For example, such tools could guide the development of treatment strategies to increase tumor-targeting and minimize off-target accumulation and associated toxicity [1,2,3,4]. The effector immune-cell populations involved in these therapeutic interventions reach peripheral tissues in relatively small numbers, in the order of a few thousand [5,6,7,8,9]. Therefore, stable labeling of immune cells and a highly sensitive imaging system with adequate tissue penetration depth for whole-body imaging are required.

For direct cell labeling, specific cell (sub)types are isolated from patients and labeled ex vivo. For example, T cells can be labeled with highly derivatized crosslinked iron oxide nanoparticles (NPs) and detected with magnetic resonance imaging (MRI) [10]. Although magnetic particle imaging (MPI) is a highly sensitive technique (detection of ∼200 cells and a resolution of ∼1 mm), there are currently no clinically available scanners [11,12,13,14]. In contrast, positron emission tomography (PET) is commonly applied in clinical practice for diagnosis, staging and response monitoring in cancer and other diseases. PET tracking of radiolabeled cells is traditionally performed with lipophilic compounds, including oxine and hexamethylpropyleneamine oxime (HMPAO), that passively diffuse across the cell membrane [15]. In clinical settings, [^111^In]In-oxine and [^99m^Tc]Tc-HMPAO are used in combination with single-photon emission computed tomography (SPECT) imaging [16,17]. Although this labeling strategy is fast and results in a good labeling efficiency, it has several drawbacks. In particular, it has been shown that cells labeled with [^111^In]In-oxine release the radionuclide up to 50% within 48 h [18], and the oxine carrier appears to be chemotoxic [19]. Furthermore, clinical SPECT systems have a low sensitivity and resolution, which hamper the tracking of low numbers of cells distributed over a large volume. In contrast, PET has a higher resolution, sensitivity and more accurate quantification compared to SPECT. New developments in PET technology hold promise for even higher sensitivity compared to current PET scanners [20]. Therefore, PET-based in vivo cell tracking is preferred over SPECT in the clinical setting. In this study, we introduced the radiometal zirconium-89 (^89^Zr) as a positron emitter, with an ideal half-life (78.41 h) for cell tracking.

More recently, [^89^Zr]Zr-oxine has been introduced for PET-based cell tracking [21,22], but it experiences similar limitations to those observed with [^111^In]In-oxine [23,24]. A multitude of ^89^Zr-labeled NPs for different applications have been reported previously [25,26,27]. For example, chitosan NPs have also been used for ^89^Zr-labeling of leukocyte cells [28]. The intrinsic labeling of the chitosan NPs was suggested to be possible via the interaction of the ^89^Zr with the OH and NH_2_ groups. High cell-labeling efficiency was achieved with up to 73% after 24 h. However, the ^89^Zr-release from the cells was also rapid, with up to 79% after 24 h. No specific activity per number of cells was reported, making it difficult to compare to other studies. In another study, in vivo tumor-associated macrophages were targeted and imaged by using ^89^Zr-desferroxiamine-NCS (DFO) conjugated dextran NPs in colon carcinoma (CT26) tumor xenograft mice [25]. Here, also some release was detected in the bones with PET images. Radiolabel release by ex vivo labeled cells is a hurdle for sensitive and specific in vivo cell tracking, as free radionuclides accumulate in off-target tissue and could lead to higher background signal and potential misinterpretation of images, while also exposing tissue to unnecessary radiation dose [29,30].

In our previous work, we have shown that poly(lactic-*co*-glycolic acid) NPs with amine groups (PLGA-NH_2_ NPs) can be used to radiolabel cells and demonstrates improved radiolabel retention compared with the oxine labeling method [31]. Here, we report the intrinsic labeling capacity of these NPs with [^89^Zr]ZrCl_4_ under various conditions. In vitro, immortalized human monocytes (THP-1) were labeled with [^89^Zr]Zr-PLGA-NH_2_ NPs and the retention of ^89^Zr in the cell was studied over time. Finally, we show that it is feasible to image ex vivo labeled THP-1 cells with PET in mice with *Staphylococcus aureus* (*S. aureus*) inflamed muscles or human breast adenocarcinoma MDA-MB-231 tumors.

## 2. Materials and Methods

### 2.1. Synthesis of Nanoparticles

The same preparation protocol was used as described before [31]. Briefly, 100 mg of poly(lactic-*co*-glycolic acid) diamine endcap copolymer (PLGA-NH_2_, Mn = 5000, Sigma-Aldrich, Merck, Saint Louis, MO, USA), 200 µL of poly(propylene glycol) (PPO, 50 mg/mL stock, Sigma-Aldrich, average Mn 2700) and 900 µL perfluoro-15-crown-5-ether (PFCE, Exfluor Inc., Round Rock, TX, USA) were dissolved in 3 mL dichloromethane (DCM, Merck, Darmstadt, Germany). Simultaneously, 500 mg of poly(vinyl alcohol) (PVA, Mw 9000–10,000 Da, 80% hydrolyzed, Sigma-Aldrich) was dissolved in 25 g of MilliQ (18.2 MΩ cm, Merck, Kenilworth, NJ, USA). The organic phase was mixed and rapidly added to the PVA solution during sonication for 3 min, at 40% amplitude, with a probe sonicator (Sonifier 250, microtip 6.4 mm, Branson Sonic Power, Saint Louis, MO, USA). The organic phase was left to evaporate overnight, at 4 °C, while stirring. After washing the NPs four times with MilliQ and snap-freezing in liquid nitrogen, the samples were lyophilized for 48 h and stored at −20 °C until needed.

### 2.2. Characterization of Nanoparticles

PLGA-NH_2_ NPs were analyzed for size, polydispersity index (PDI) and zeta potential, in the same way as in our previous study [31]. The NPs were dissolved at 0.1 mg/mL in MilliQ, and both size and PDI were measured by using a NANO-flex (Microtrac, Inc., Duesseldorf, Germany), and the data were analyzed by using Microtrac software (Microtrac FLEX 11.1.0.2, Duesseldorf, Germany). The zeta potential was measured by using Zetasizer Nano ZS (Malvern Instruments, Worcestershire, United Kingdom), where similar NP concentrations were dissolved in NaCl (5 mM, pH 7.4). Encapsulation efficiency of PFCE was measured by using a nuclear magnetic resonance (NMR, Bruker Avance III 400 MHz, Bruker BioSpin, Ettlingen, Germany) spectrometer coupled with a Broad Band Fluorine Observation (BBFO) probe. NPs, ~5 mg, were dissolved in 500 µL deuterium oxide (D_2_O) containing 100 µL 1 volume% trifluoroacetic acid (TFA) in D2O. For quantification, the interscan relaxation delay (D1) was set at 5 times the relaxation time (T1) of TFA, at 20 s. The data were evaluated with Mestrenova 10.0.2 (Mestrelab Research, Escandido, CA, USA).

### 2.3. Characterization of Nanoparticles Stability in Human Serum and PBS over Time

PLGA-NH_2_ and Zr-PLGA-NH_2_ NPs’ size and PDI were measured in 100% human serum (human male AB plasma, Sigma-Aldrich, USA) and PBS at 0, 1, 2, 4, 6, 24, 48, 72, 168 and 336 h. First, the NPs were labeled with non-radioactive zirconium (932 µg zirconium/mg NP in 0.05 M HCl, pH 1.1–1.4, MO, USA) in metal-free 0.5 M ammonium acetate (NH_4_Ac, pH 5.5), which is similar to ^89^Zr-labeling (see Zirconium-89 labeling of PLGA and PLGA-NH_2_ NPs). Second, both PLGA and Zr-PLGA-NH_2_ NPs were dissolved at a concentration of 10 mg/mL in PBS or 100% human serum. The samples were incubated at 37 °C, in a thermomixer, for the indicated timepoints. Last, 10 µL of NP solution was transferred to 990 µL MilliQ (0.1 mg/mL), and both size and PDI were measured as explained above.

### 2.4. [^89^Zr]ZrCl_4_ Preparation from ^89^Zr-Oxalate

In order to obtain [^89^Zr]ZrCl_4_, we removed oxalate by using a Sep-Pak Light Accell Plus QMA Cartridge (Waters, Dublin, Ireland). The Sep-Pak was activated with 10 mL acetonitrile and then washed with 10 mL 0.9% NaCl, 10 mL 1 M HCl and 10 mL water. [^89^Zr]Zr-oxalate (Cyclotron VU, Amsterdam, The Netherlands) was added, and the cartridge was washed with 50 mL water. Finally, the ^89^Zr-label was eluted with 1 mL HCl (0.1 M) in 100 µL aliquots.

### 2.5. Intrinsic ^89^Zr-Labeling of PLGA and PLGA-NH_2_ NPs

This experiment was performed in the same manner as described in our previous study [31]. For intrinsic labeling, 1 mg NPs were dissolved in 0.5 M NH_4_Ac and incubated with 1–5 MBq [^89^Zr]ZrCl_4_, at 37 °C, for 30 min. After washing the NPs 3 times with PBS, the labeling efficiency and radiochemical purity were determined with instant Thin-Layer Chromatography (iTLC). Labeling efficiency was calculated as the fraction of radioactivity at the origin to the total amount of radioactivity. Unless otherwise stated, the NPs were washed until a radiochemical purity of >95% was obtained. All radioactive labeling was performed in 0.5 M NH_4_Ac, pH 5.5, unless stated otherwise.

### 2.6. ^89^Zr-Retention of PLGA-NH_2_ NPs in PBS and Human Serum

[^89^Zr]Zr-PLGA-NH_2_ NPs (1–5 MBq/mg, 10 mg/mL) were incubated in 100% human serum and PBS, at 37 °C, for 2 weeks. The ^89^Zr-retention was measured at 0, 1, 2, 4, 6, 24, 48, 72 and 336 h after incubation with iTLC.

### 2.7. EDTA Challenge

[^89^Zr]Zr-PLGA-NH_2_ NPs (3 MBq/mg, 10 mg/mL) were challenged with 0.1, 1, 10 and 50 mM EDTA (corresponds to approximately 0.1, 1, 10 and 50 equivalents more EDTA to NP) in PBS at 37 °C for 2 weeks. At 0, 1, 2, 4, 6, 24, 48, 72, 168 and 336 h, samples of 1 µL were analyzed with iTLC.

### 2.8. Cell Culture

The immortalized human monocyte cell line THP-1 (ATCC^®^ TIB-202^TM^, VA, Gaithersburg, MD, USA) was used for cell labeling (passage of <20). The cells were maintained in culture as described previously [31].

The human adenocarcinoma cell line (MDA-MB-231, passage 46, ATCC^®^ HTB-26^TM^, Gaithersburg, MD, USA) was cultured under the same conditions.

### 2.9. [^89^Zr]Zr-PLGA-NH_2_ NP Labeling of THP-1 Cell Line and Retention of Radiolabel over Time

THP-1 cells were incubated with [^89^Zr]Zr-PLGA-NH_2_ NPs, at a concentration of 7.5 ± 0.3 MBq/1 mg NP/10^6^ cells, at 37 °C, for 2 h. As a control, we treated the THP-1 cells in the same manner, without the addition of [^89^Zr]Zr-PLGA-NH_2_ NPs, but with PBS. Subsequently, cells were washed to remove NPs which were not taken up by the cells. After labeling and washing, cells were incubated at culture conditions for 1, 2, 4, 6, 24 and 48 h. At every timepoint, the cells were first measured for radioactivity for 1 min with a γ-counter (wizard 2480 Automatic Gamma Counter, PerkinElmer, Downers Grove, IL, USA). The cells were then centrifuged at 300× *g* for 5 min, the supernatant was removed and the cells were resuspended in fresh PBS before another radioactivity measurement. The percentage retained radioactivity in the cells was calculated by dividing the activity measured after removal of supernatant by total amount of radioactivity before centrifugation, multiplied by 100.

### 2.10. Cell Counting

Cell numbers after an experiment were counted with Luna-II Automated Cell Counter (Logos Biosystems, Inc., Anyang, South Korea). The mixture of cells with trypan blue (1:1) was transferred to a counting cassette (Logos Biosystems, Inc., Korea) before automated counting. Living cells were used for calculating the specific activity per number of cells by dividing the total activity associated with the pellet with the number of living cells times hundred.

### 2.11. CellTiter-Glo Assay

For ATP content measurement, 80,000 cells were diluted with PBS to a volume of 350 µL and mixed with 350 µL of premixed substrate and buffer CellTiter-Glo (Promega, Madison, WI, USA). After a short vortex, the samples were incubated for 10 min, at room temperature (RT). From each sample, 200 µL in triplicate was transferred to a 96-wells plate (black flat bottom), and luminescence was measured by using a Tecan Infinite M200 PRO and software Tecan i-control (attenuation automatic, integration time 1000 millisecond, settle time 0 millisecond, Tecan, Grödig, Austria). Controls were set to 100%, and sample results were compared to this.

### 2.12. Animal Experiments

For animal experiments, the guidelines set by the Nijmegen and European Animal Experiments Committee (CCD application 2018-0011 and 2020-0007) were followed. The animals were housed in groups in individually ventilated Blue line cages. To determine [^89^Zr]Zr-PLGA-NH_2_ NPs biodistribution and blood clearance, 6 female C57BL/6JRj mice (Janvier Labs) were used (age 6–8 weeks, weight 18.4 ± 1.2 g). For PET and MRI studies with [^89^Zr]Zr-PLGA-NH_2_ NPs labeled THP-1 cells in Matrigel, 12 female BALB/cAnNRj-Foxn1nu/Foxn1nu mice (Janvier Labs) were used (age 6–8 weeks, weight 20.0 ± 0.9 g). In vivo tracking of [^89^Zr]Zr-THP-1 cells in *S. aureus* and MDA-MB-231 tumor models were performed in 11 female BALB/CAnN.Cg-Foxn1nu/Crl mice (Charles River) (age 6–8 weeks, weight 16.5 ± 2.3 g). The mice were allowed to acclimate for 1 week before the start of the experiments. Upon arrival, the mice were randomly identified with tattoos by biotechnicians who were blinded to the experimental setup.

### 2.13. [^89^Zr]Zr-PLGA-NH_2_ NPs Biodistribution and Blood Clearance in C57BL/6JRj Mice

At day 0, all mice were i.v. injected via the tail vein with 200 µL PBS containing 1.81 ± 0.61 MBq/5 mg [^89^Zr]Zr-PLGA-NH_2_ NPs (the particles were washed until <5% release of free ^89^Zr was measured compared to previous washing step). For blood kinetics, blood samples were collected via saphenous vein or heart puncture (when sacrificed), at 30 min (3 mice), 1 h (6 mice), 2 h (3 mice), 4 h (6 mice), 24 h (6 mice), day 2 (6 mice), day 3 (6 mice), day 7 (3 mice) and day 14 (3 mice). For ex vivo biodistribution, organs (spleen, liver, kidney, heart, lungs, pancreas, bladder, duodenum, ileum, colon, brain, muscle, lymph nodes (LN, inguinal), femur, bone marrow, thymus, brown fat, stomach, salivary glands and knees) were harvested after euthanizing the mice with CO_2_/O_2_ asphyxiation at day 3 (3 mice) and day 14 (3 mice). Radioactivity of the blood and organs was measured for 1 min with a γ-counter. Tissue and blood were measured for % injected dose per gram (%ID/g) with simultaneous measurements of aliquots from injected fluid. GraphPad Prism (GraphPad Software Inc., San Diego, CA, USA) was used for calculating the blood half-life (t_1/2_) with nonlinear regression with one-phase decay, as computed by using the formula In(2)/K, with K as a rate constant, expressed in reciprocal of the *x*-axis time units.

For in vivo PET/MRI imaging, mice were anesthetized by using isoflurane in oxygen (33% oxygen + 67% air) before imaging. Isoflurane (5%) was used for induction and maintained at 1–2% during scanning. Mice were imaged at 1 h (3 mice), 4 h (3 mice), 24 h (3 mice), day 3 (3 mice), day 7 (3 mice) and day 14 (3 mice) with PET (Siemens Preclinical Solution) and followed immediately by MRI (Bruker ClinScan 70/30 7T, Bruker BioSpin, Ettlingen, Germany). During scanning, the temperature of the mice was maintained at 37 °C, using a heating bed. For reconstruction of PET scans (30–60 min), Inveon Acquisition Workspace software (version 1.5, Siemens Preclinical Solution, Erlangen, Germany) was used with the reconstruction algorithm OSEM3D/SO-MAP and the following settings: no scatter correction; using scale factor, 0; matrix, 256 × 256; image zoom, 1; frames, all; OSEM Iterations, 2; MAP Iterations, 18; target resolution, 1.5 mm; voxel size, 0.4 × 0.4 × 0.8 mm.

Immediately following the PET scan, mice were transported by using the same imaging bed to the MRI scanner for anatomical imaging, where they were scanned for a duration of 32 min. A birdcage body coil (Bruker BioSpin, Ettlingen, Germany) with 86 mm inner diameter was used for image acquisition. After a localizer scan, the following settings were used for the 3D gradient echo scans: acquisition time = 32 min; repetition time = 30 ms; echo time = 1.3 ms; flip angle = 15 degrees; field of view = 120 × 56 × 32 mm; and matrix = 576 × 270 × 160, resulting in an isotropic resolution of 0.20 mm^3^.

The PET/MRI images were merged, and Inveon Research Workplace software (version 4.1) was used to create maximum-intensity projections. For the overlay, a reference tube with ^89^Zr in PBS was placed on the scan bed.

### 2.14. PET and MRI Imaging of [^89^Zr]Zr-PLGA-NH_2_ NPs Labeled THP-1 Cells in Matrigel

At day 0, 10,000 (395 ± 179 Bq, *n* = 4) or 100,000 (3950 ± 1790 Bq, *n* = 4) of [^89^Zr]Zr-THP-1 cells suspended in PBS were mixed with Matrigel (2:1 (*v/v*) PBS:Matrigel, BD Matrigel Matrix Basement Membrane (20.20 MG/ML), BD Biosciences, Bedford, MA, USA) before 200 µL was s.c. injected in the flank lower part of the back and abdomen. In addition, a mixture of Matrigel (1:1) and 1.56 ± 0.47 µg [^89^Zr]Zr-PLGA-NH_2_ NPs (3400 ± 2194 Bq, *n* = 4) in PBS was s.c. injected as a control. For blood kinetics, blood samples were collected via saphenous vein or heart puncture (after sacrifice) at 30 min (4 mice), 1 h (4 mice), 4 h (4 mice) and 24 h (6 mice). For ex vivo biodistribution, organs and Matrigel were harvested and measured as described previously.

For in vivo PET/MRI imaging, mice were imaged at 1 h (4 mice) and 24 h (4 mice) with PET and followed immediately by MRI (Bruker BioSpec 117/16 11.7T, Bruker BioSpin, Ettlingen, Germany). For PET, the same settings were used as previously described.

Immediately following the PET scan, mice were transported by using the same imaging bed to the MRI scanner for anatomical imaging, where they were scanned for a duration of 45 min. A birdcage body coil (Bruker BioSpin, Ettlingen, Germany) with 72 mm inner diameter was used for image acquisition. After a localizer scan, the following settings were used for the 3D gradient echo scans: acquisition time = 42 min; repetition time = 100 ms; echo time = 2.2 ms; flip angle = 30 degrees; field of view = 100 × 40 × 40 mm; and matrix = 400 × 160 × 160, resulting in an isotropic resolution of 0.25 mm^3^.

### 2.15. In Vivo Tracking of [^89^Zr]Zr-THP-1 Cells in S. aureus and MDA-MB-231 Tumor Models

*S. aureus* was purchased from QM Diagnostics (1.05 × 10^9^ Colony-Forming Units (CFU)/mL PBS). At day −2, *S. aureus* was mixed (1:1) with freshly extracted blood from donor mice, and 50 µL of the mixture was injected intramuscular in the right hind leg of mice (*n* = 5). In addition, PBS was mixed (1:1) with donor blood, and 50 µL was injected in the contralateral hind leg muscle of the same mice. At day 0, [^89^Zr]Zr-THP-1 cells (5.8 × 10^6^ cell/mouse, 90.75 ± 12.84 kBq/mouse) were injected i.v. via the tail vein.

For the tumor model, MDA-MB-231 cells (5 × 10^6^ cells/mouse) were suspended in RPMI-1640 medium, mixed with Matrigel (1:1) and injected s.c. in the right hind back side at day −22 (*n* = 5, 200 µL/mouse). When tumors reached a size of approximately 0.1 cm^3^, [^89^Zr]Zr-THP-1 cells were injected i.v. via the tail vein (5 × 10^6^ cells/mouse, 164.80 ± 12.99 kBq/mouse).

For PET/MRI imaging and processing of images, the same settings were used as in PET and MRI imaging of [^89^Zr]Zr-PLGA-NH_2_ NPs labeled THP-1 cells in Matrigel.

### 2.16. Statistical Analysis

Statistical analysis was performed in GraphPad Prism software. A Gaussian distribution with a two-tailed unpaired Student’s *t*-test was used to analyze differences in two groups. A *p*-value of <0.05 was considered statistically significant. When comparing three or more groups, a one-way ANOVA test with a Tukey test correction of multiple comparisons was performed. A two-way ANOVA test with a Sidak correction of multiple comparisons was used in the biodistribution experiments (Figure 2A), while a Tukey’s correction was used to compare the groups in the Matrigel experiments (Figure 5A). All comparisons were based on a minimum of three replicates.

## 3. Results

### 3.1. Characterization of Particles

The PLGA-NH_2_ particles, prepared as described previously [31], were labeled with non-radioactive zirconium (Zr) and characterized for diameter, polydispersity (PDI) and zeta potential (Table 1). The diameter increased slightly when the PLGA-NH_2_ NPs were labeled with zirconium from 189 ± 1.9 nm to 196 ± 4.1 nm. The PDI remained the same while the zeta potential increased to a more neutral charge, from −2.3 ± 0.9 mV to −0.3 ± 0.4 mV. These results indicate that NP characteristics were altered by the Zr-labeling, with increased size and zeta potential.

### 3.2. Stability of the PLGA-NH_2_ and Zr-PLGA-NH_2_ NPs in PBS and Human Serum

To assess whether the characteristics of the particles changed over time, we incubated the PLGA-NH_2_ and Zr-PLGA-NH_2_ NPs in PBS and 100% human serum at 37 °C for a period of 2 weeks. The diameter of the NPs remained stable (~200 nm) in PBS for 72 h and was increased (~300 nm) at 336 h (Appendix A). Similarly, the PDI of both NPs remained stable (~0.08) for 72 h and was increased (~0.2) at 336 h. In human serum, the diameter of the NPs was increased (>200 nm) at 336 h (Appendix A). The PDI of both samples showed similar fluctuations as observed for the diameter over time.

### 3.3. [^89^Zr]ZrCl_4_ Labeling of PLGA and PLGA-NH_2_ NPs

PLGA and PLGA-NH_2_ NPs were radiolabeled with [^89^Zr]ZrCl_4_, where a labeling efficiency of 7.1 ± 0.9% and 101.5 ± 1.1% for PLGA NPs and PLGA-NH_2_ NPs (*p* < 0.0001, Figure 1A) was observed, respectively, showing efficient ^89^Zr-labeling of PLGA-NH_2_ NPs, without the need for additional chelator. To evaluate the effect of buffer on labeling efficiency, the PLGA-NH_2_ NPs were labeled in 0.5 M HEPES, MES and NH_4_Ac buffer at a pH of 5.5 (Figure 1B). Labeling efficiency was highest for the NH_4_Ac buffer (76 ± 2%, *p* < 0.0001 compared to HEPES and MES buffers). We therefore continued to label PLGA-NH_2_ NPs in NH_4_Ac buffer. The retention of the ^89^Zr by the NPs was measured in PBS and 100% human serum. In PBS and 100% human serum, the ^89^Zr-retention was ∼85 ± 15% after 336 h (Figure 1C). In addition, [^89^Zr]Zr-PLGA-NH_2_ NPs were challenged with EDTA at 37 °C, for 336 h. After an initial release of ^89^Zr from the NPs during the first 6 h, a gradual and EDTA concentration-dependent release of ^89^Zr was observed for up to 336 h (Figure 1D). From these results, we can conclude that ^89^Zr was interacting with the PLGA-NH_2_ NPs and retained by the NPs in PBS and human serum. However, the ^89^Zr-label could be challenged by EDTA.

### 3.4. In Vivo Biodistribution of [^89^Zr]Zr-PLGA-NH_2_ NPs in C57BL/6 Mice

The in vivo biodistribution of [^89^Zr]Zr-PLGA-NH_2_ NPs upon intravenous (i.v.) injection was evaluated in C57BL/6 mice. The concentration of [^89^Zr]Zr-PLGA-NH_2_ NPs in blood decreased rapidly, and the calculated blood half-life (t_1/2_) was 28 ± 6 min (Figure 2A and Appendix A).

Spleen, liver and bone marrow were the main organs for NP accumulation, as demonstrated by the result of the ex vivo measurements and PET/MRI scans (Figure 2B and Figure 3 and Appendix A). In addition, we also observed accumulation in femur (5.9 ± 0.1% ID/g at day 14) and knees (7.2 ± 1.8% ID/g at day 14). Taken together, these results show that the particles are cleared from the blood in the first 24 h after injection and that the spleen, liver and bone marrow are the main accumulation sites.

### 3.5. [^89^Zr]Zr-PLGA-NH_2_ NPs Labeling of THP-1 Cells and Retention over Time

THP-1 cells, immortalized human monocytes, were labeled with [^89^Zr]Zr-PLGA-NH_2_ NPs ([^89^Zr]Zr-THP-1 cells), where a labeling efficiency of 4.03 ± 0.16% was observed, resulting in a specific activity of 279 ± 10 kBq/10^6^ cells. The [^89^Zr]Zr-THP-1 cells retained 79.6 ± 5.9% of the radiolabel at 48 h after incubation (Figure 4A). Cell counting showed that 76.4 ± 15.2% of [^89^Zr]Zr-THP-1 cells remained alive over 48 h, while the ATP content, as measured with CellTiter-Glo assay in the cells, did not decrease (119.7 ± 9.4% compared with control samples; Figure 4B,C). In summary, [^89^Zr]Zr-PLGA-NH_2_ NPs could stably label THP-1 cells, which remained viable over 48 h.

### 3.6. [^89^Zr]Zr-THP-1 Cells for In Vivo PET/MRI Imaging

To determine PET sensitivity for the detection of low numbers of [^89^Zr]Zr-THP-1 cells, three groups of mice were injected subcutaneously (s.c.) with Matrigel containing 10,000 [^89^Zr]Zr-THP-1 cells, 100,000 [^89^Zr]Zr-THP-1 cells or [^89^Zr]Zr-PLGA-NH_2_ NPs (Figure 5). All three Matrigel depositions were visible on the PET scans (Figure 6). From the biodistribution data, we can see that the blood and organ signals were low, indicating that the radioactive signal remains at the Matrigel for over 24 h (Figure 5 and Appendix A).

### 3.7. In Vivo PET/MRI Tracking of [^89^Zr]Zr-THP-1 Cells in Local S. aureus Infection and MDA-MB-231 Tumor Model

[^89^Zr]Zr-THP-1 cells were tracked in vivo in a local intramuscular *S. aureus* infection and MDA-MB-231 tumor model. For the *S. aureus* model, i.v. injected [^89^Zr]Zr-THP-1 cells were rapidly cleared from blood within 30 min. Ex vivo biodistribution study at 24 h showed that the organs with the highest uptake were spleen, liver, lung and bone marrow (Figure 7A,B and Appendix A). Furthermore, 0.46 ± 0.10% of the total administered [^89^Zr]Zr-THP-1 cells (corresponding to 26.89 ± 5.79 × 10^3^ cells) accumulated in the *S. aureus*–infected muscle, compared with 0.04 ± 0.02% (2.14 ± 0.89 × 10^3^ cells) in the control muscle. A PET signal was detected as early as 4 h post-injection in the *S. aureus*–infected muscle, which increased at 24 h post-injection (Figure 8 and Appendix A).

In MDA-MB-231 tumor-bearing mice, [^89^Zr]Zr-THP-1 cells were also cleared from blood within 30 min. The spleen, liver, lung and bone marrow showed a high uptake of the radiolabeled cells at 24 h (Figure 7C,D and Appendix A). The number of [^89^Zr]Zr-THP-1 cells accumulated at the tumor site was 0.11 ± 0.05% (corresponding to 5.55 ± 2.53 × 10^3^ cells) of the total injected cells, compared with 0.02 ± 0.00% (0.96 ± 0.19 × 10^3^ cells) in the muscle of the foreleg. Moreover, a low PET signal in the tumor was detected at 4 h, which slightly increased at 24 h post-injection (Figure 8 and Appendix A).

Together, these sets of experiments show that it was feasible to track ex vivo [^89^Zr]Zr-PLGA-NH_2_ NPs labeled THP-1 cells in two disease models.

## 4. Discussion

PET is currently the most sensitive whole-body-imaging modality for clinical studies that is ideal for in vivo tracking of small numbers of labeled cells. The long-lived positron emitter ^89^Zr^4+^ allows for imaging up to several days post-injection. This prompted us to explore the potential of [^89^Zr]Zr-PLGA-NH_2_ NPs for cell labeling and in vivo tracking with PET.

We previously developed PLGA-NH_2_-based NPs that were able to intrinsically complex and retain [^111^In]InCl_3_ for SPECT [31]. Here we demonstrated that these NPs also allow for intrinsic labeling with [^89^Zr]ZrCl_4_ for PET. As expected, labeling with non-radioactive Zr slightly increased the NPs’ size and zeta potential.

PLGA-NH_2_ NPs showed efficient labeling with [^89^Zr]ZrCl_4_, compared to normal PLGA NPs without -NH_2_. In PBS and human serum, ^89^Zr was retained for >80% by the NPs for up to 2 weeks. This indicates that the particles are able to retain the ^89^Zr-label without the use of a chelator, such as desferrioxamine (DFO). However, when challenged with EDTA, ^89^Zr was partly released from the particles, even at 0.1 mM (0.1 equivalents of EDTA) concentration. ^89^Zr-release upon EDTA (1000 equivalents) challenge was also reported for DFO-conjugated trastuzumab, which showed a release of 25% and 50% in the first 24 h 7 days, respectively, which is slower than observed in our study [32]. From the literature, it was known that ^89^Zr requires a strong Lewis base, such as OH^−^ ions, and an 8-coordination for optimal binding and retention [33], which cannot be secured in the NPs, as chelation depends on free primary amine groups. However, for our application, the [^89^Zr]Zr-PLGA-NH_2_ NPs mainly serve the purpose of ex vivo cell labeling, and the release, in the first instance, is mainly limited to the intracellular compartments of the labeled cells. However, in the course of time or upon cell death, ^89^Zr can be released and redistributed within the body.

The biodistribution of the [^89^Zr]Zr-PLGA-NH_2_ NPs was in line with our previous observations with [^111^In]In-PLGA-NH_2_ NPs [34]. The signal at the tail was probably due to partial s.c. injection of the NPs. Interestingly, the accumulation in liver was half that of [^111^In]In-PLGA-NH_2_ NPs [31]. Furthermore, in spleen, activity at day 14 was only 50 %ID/g for [^89^Zr]Zr-PLGA-NH_2_ NPs, while it was >100 %ID/g for [^111^In]In-PLGA-NH_2_ NPs. Accumulation of ^89^Zr was observed in the femur and knee at day 3, but this did not increase further at day 14. From the literature, it is known that free ^89^Zr released from the targeting vehicle has the tendency to accumulate in bone tissue [29]. The radioactivity in femur and knee might be explained by (I) the <5% free ^89^Zr present during injection of the NPs, (II) ^89^Zr-release from the NPs after injection or (II) macrophages and monocytes that take up the NPs and are present in or migrate to bone marrow.

The labeling of the THP-1 cells with [^89^Zr]Zr-PLGA-NH_2_ NPs was not very efficient, as only ∼4% of the NPs was taken up by the cells. In general, cell labeling with [^89^Zr]Zr-oxine is faster (15–30 min) and more efficient (10–40% labeling efficiency) when compared with NP-based cell labeling [35,36,37,38]. However, the specific activity of the NPs labeled cells was in range with the results from the literature, where human mesenchymal stem cells or chimeric antigen receptor (CAR) T cells were labeled for in vivo imaging with a broad range of specific activity of 0.009–0.370 MBq/10^6^ cells, using desferrioxamine or oxine as carrier [21,37,39,40]. Moreover, higher specific activity per cell is not desired, as this could lead to radiotoxicity [37]. Furthermore, ^89^Zr was retained by the cells up to 48 h after incubation, which was comparable to [^111^In]In-PLGA-NH_2_-labeled moDC cells. Different type of cells (for example, CAR T cell and natural killer cells) labeled with [^89^Zr]Zr-oxine showed a similar decrease of radioactivity over a period of 48 h [22,37,41]. The ^89^Zr release from [^89^Zr]Zr-oxine-labeled cells was also rapid for certain cell types (DCs and CAR T cells), i.e., >25% release after 2 days. These indicate that the NPs used in this study could play a role in cell labeling and in vivo tracking. However, future studies are needed to demonstrate feasibility of radiolabeling of other cell types, such as T cells. One strategy to enhance overall cellular uptake would be to modify the coating of NPs with, for example, cell-penetrating peptides or Lipofectamine [42,43,44]. Alternatively, to improve labeling of specific subsets of immune cells, NPs can be decorated with antibodies or peptides with the desired specificity [45,46].

In vivo studies showed that we were able to detect small numbers of labeled THP-1 cells, using PET. A clear signal was observed in mice which were transplanted s.c. with 10,000–100,000 [^89^Zr]Zr-THP-1 cells (395–3950 Bq). Furthermore, minimal redistribution of radioactivity to other organs was observed, except for the femur and bone marrow, potentially caused by [^89^Zr]Zr-THP-1 cells migrating to bone marrow or ^89^Zr released from the cells. This indicates that ^89^Zr is well retained inside cells.

Next, we injected [^89^Zr]Zr-THP-1 cells i.v. and tracked their biodistribution in *S. aureus* inflammation model and a MDA-MB-231 tumor model. We detected a radioactive signal in the inflamed muscle and at the tumor site. However, it should be noted that the tumor accumulation was minimal, most likely because the tumor environment is less chemotactic compared with the *S. aureus* induced inflammation. Other studies have also developed techniques for PET-based cell tracking. For example, [^89^Zr]Zr-oxine-based cell labeling has been evaluated in several studies with different type of cells and disease models. Recently, the potential of surface labeling with [^89^Zr]Zr-DFO was shown by using human cardiopoietic stem cells for in vivo tracking in an ischemic-heart-failure mice model. Alternatively, a signal cell labeling and tracking was demonstrated with [^68^Ga]Ga-mesoporous silica NPs, using PET [47]. The concept of single-cell tracking is highly challenging, as a high load of radioactivity per cell (>70 Bq) is required for accurate tracking. This could pose a problem in prolonged studies (24–72 h), since more radioactivity per cell would be required, as the half-life of ^68^Ga is 67 min. Single-cell tracking would be interesting to study the behavior of that single cell; however, most effector mechanisms require cooperation with a multitude of other cells [48].

## 5. Conclusions

As PET is a highly sensitive imaging modality, in combination with novel cell-labeling approaches, it is ideally positioned for whole-body in vivo cell tracking. Here we expanded on our previous radiolabeling strategy and demonstrated for the first time that [^89^Zr]Zr-PLGA-NH_2_ NPs can be used as a tool for cell labeling and sensitive in vivo cell tracking, using PET. For future (clinical) applications, however, cell-labeling efficiency can be improved by coating the surface of the NPs with cell-specific antibodies, peptides, nanobodies or other targeting agents.

## Figures and Tables

**Figure 1 cancers-13-05069-f001:**
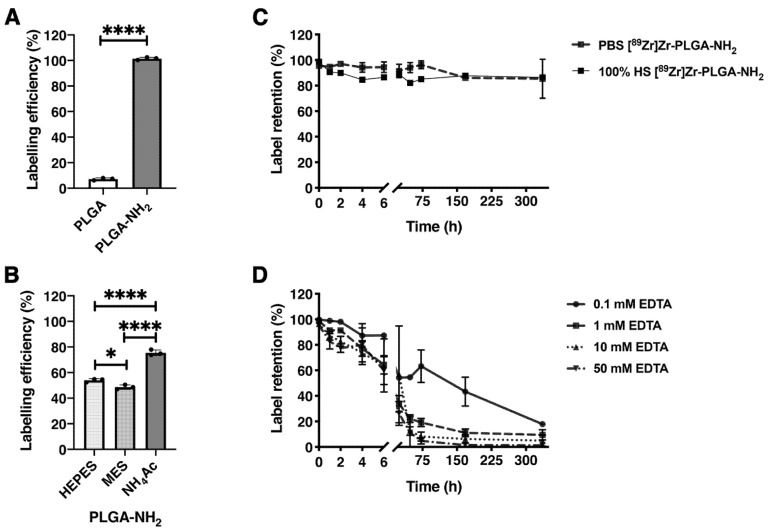
^89^Zr-labeling of NPs and label retention. (**A**) Labeling efficiency of PLGA and PLGA-NH_2_ NPs with [^89^Zr]ZrCl_4_ (*n* = 3). (**B**) ^89^Zr-labeling of PLGA-NH_2_ NPs in 0.5 M and pH 5.5 HEPES, MES and NH_4_Ac labeling buffers (*n* = 3). (**C**) ^89^Zr-retention by PLGA-NH_2_ NPs was examined in PBS and 100% human serum (100% HS) at 37 °C at 0, 1, 2, 4, 6, 24, 48, 72, 168 and 336 h (*n* = 3). (**D**) EDTA concentration range challenge was performed at 37 °C at 0, 1, 2, 4, 6, 24, 48, 72, 168 and 336 h (*n* = 3). * *p* = 0.0276, **** *p* < 0.0001.

**Figure 2 cancers-13-05069-f002:**
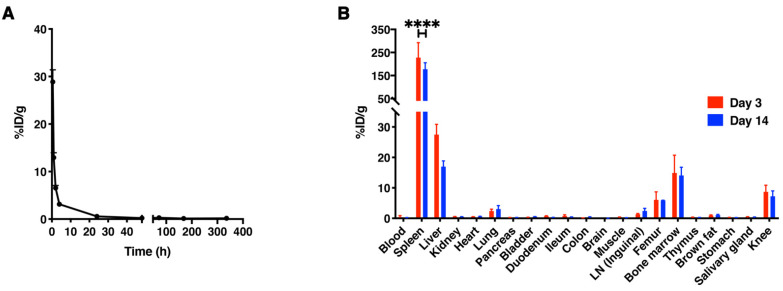
Blood clearance and biodistribution of [^89^Zr]Zr-PLGA-NH_2_ NPs. (**A**) [^89^Zr]Zr-PLGA-NH_2_ NPs clearance from blood after intravenously injection in C57BL/6 mice, measured at 0.5, 1, 2, 4, 6, 24, 48, 72, 168 and 336 h (*n* = 3–6). (**B**) Organ accumulation of the [^89^Zr]Zr-PLGA-NH_2_ NPs at day 3 and day 14 post-injection (*n* = 3 per group). Abbreviations: %ID/g, % injected dose per gram of organ; LN, lymph node. **** *p* < 0.0001.

**Figure 3 cancers-13-05069-f003:**
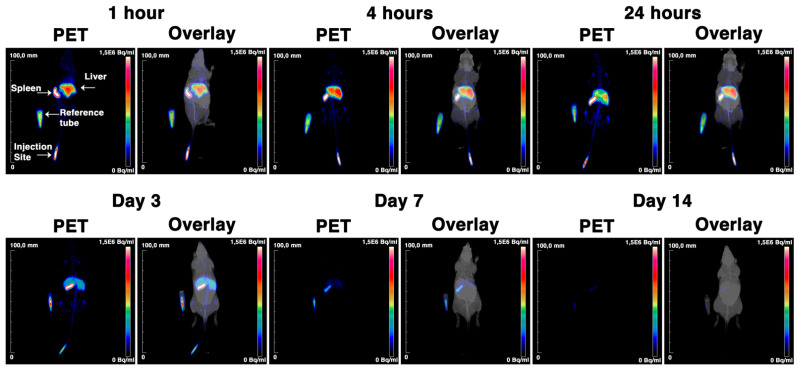
PET/MRI images of [^89^Zr]Zr-PLGA-NH_2_ NPs in C57BL/6 mice. C57BL/6JRj mice were intravenously injected with [^89^Zr]Zr-PLGA-NH_2_ NPs and imaged with PET/MRI at 1 h, 4 h, 24 h, 3 days, 7 days and 14 days post-injection. The reference tube contains 10% of the injected ^89^Zr dose.

**Figure 4 cancers-13-05069-f004:**
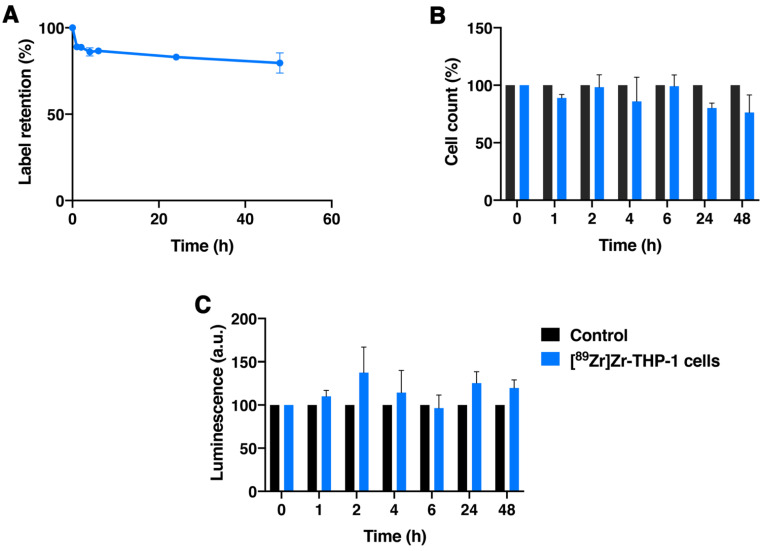
THP-1 labeling and retention of radionuclide over time. The ^89^Zr-retention by THP-1 cells was measured for 1, 2, 4, 6, 24 and 48 h, at culture conditions; (**A**) the cells were measured for relative radioactivity after one spin; (**B**) viable cell numbers counted with trypan blue staining; and (**C**) the ATP content of cells as a measure with CellTiter-Glo for cell viability. In all experiments, controls are THP-1 cells which were treated in the same way as other conditions without [89Zr]Zr-PLGA-NH2 but with PBS. Moreover, the controls did not change in value over time and therefore were set to 100%, and the remaining samples were compared to the controls. The mean and standard deviation of at least three independent experimental datasets are shown.

**Figure 5 cancers-13-05069-f005:**
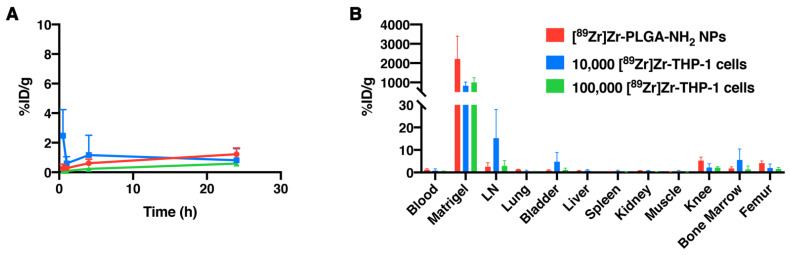
Blood values and biodistribution of [^89^Zr]Zr-THP-1 cells in BALB/cAnNRj-Foxn1nu/Foxn1nu mice. (**A**) Blood values of [^89^Zr]Zr-PLGA-NH_2_ NPs (*n* = 4), 10,000 [^89^Zr]Zr-THP-1 cells (*n* = 4) and 100,000 [^89^Zr]Zr-THP-1 cells (*n* = 4) after subcutaneous injection in BALB/cAnNRj-Foxn1nu/Foxn1nu mice, measured at 0.5, 1, 4 and 24 h post-injection. (**B**) Matrigel and organ accumulation of the [89Zr]Zr-PLGA-NH2 NPs, 10,000 [89Zr]Zr-THP-1 cells and 100,000 [89Zr]Zr-THP-1 cells in the same mice. Abbreviations: %ID/g, % injected dose per gram of organ; LN, lymph node.

**Figure 6 cancers-13-05069-f006:**
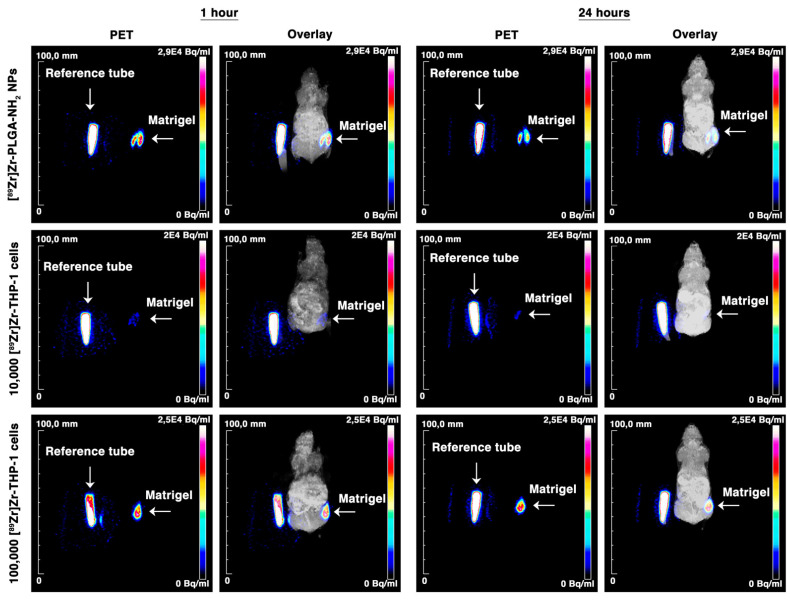
PET and MRI images of subcutaneous injected [^89^Zr]Zr-THP-1 cells in BALB/cAnNRj-Foxn1nu/Foxn1nu mice. Mice were subcutaneously injected with Matrigel containing [^89^Zr]Zr-PLGA-NH_2_, 10,000 [^89^Zr]Zr-THP-1 cells or 100,000 [^89^Zr]Zr-THP-1 cells. PET and MRI scans were performed after 1 and 24 h. The reference tube contains 10% of the injected dose. Brightness (+100 points) and contrast (+70 points) were increased for better visualization, with Adobe Photoshop, in all the images.

**Figure 7 cancers-13-05069-f007:**
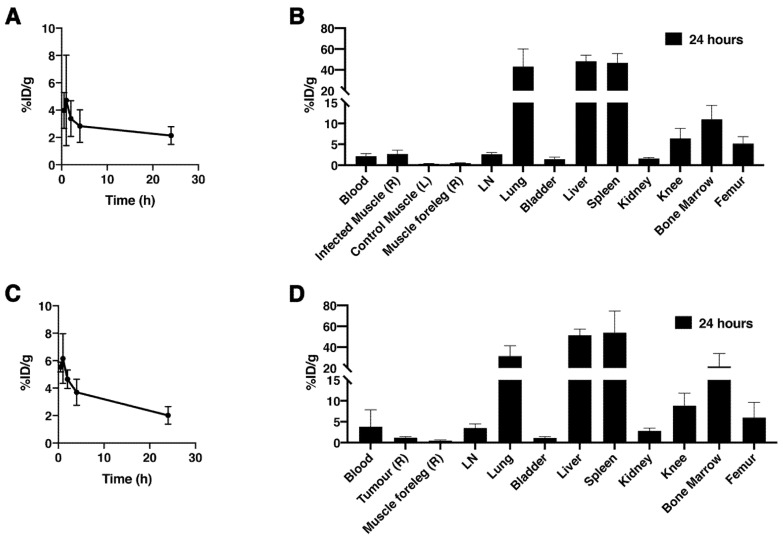
Biodistribution and blood clearance of [89Zr]Zr-THP-1 cells in *Staphylococcus aureus* (*S. aureus*) and MDA-MB-231 tumor models. Blood clearance and organs were measured after injection of the ex vivo labeled [^89^Zr]Zr-THP-1 cells in *S. aureus* (**A**,**B**) and MDA-MB-231 tumor (**C**,**D**) BALB/CAnN.Cg-Foxn1nu mice models. Blood was collected at 0.5, 1, 2, 4 and 24 h (*n* = 3–5). Organs were harvested after 24 h (*n* = 4–5). Additional information: infected muscle (R), *S. aureus* + blood (1:1) injected in right hind leg; control muscle, PBS + blood (1:1) injected in left hind leg; muscle foreleg (R), negative control muscle without any injection. Abbreviations: %ID/g, % injected dose per gram of organ; LN, lymph node.

**Figure 8 cancers-13-05069-f008:**
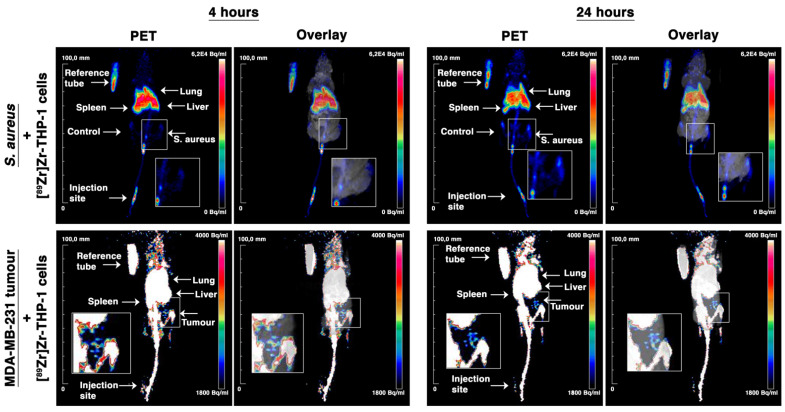
In vivo PET/MRI tracking of [89Zr]Zr-THP-1 cells in *Staphylococcus aureus* (*S. aureus*) and MDA-MB-231 tumor models. *S. aureus* or MDA-MB-231 tumor cells were injected intramuscular or subcutaneously injected in BALB/CAnN.Cg-Foxn1nu mice, respectively. Subsequently, ex vivo 89Zr-labeled THP-1 cells ([89Zr]Zr-THP-1, 5–5.8 × 10^6^ cells/mouse) were injected intravenously and followed with PET/MRI for 24 h.

**Table 1 cancers-13-05069-t001:** Characterization of PLGA-NH_2_ and Zr-PLGA-NH_2_ NPs: size (*n* = 3), PDI (*n* = 3) and zeta potential (*n* = 3).

Sample	Diameter (nm)	Polydispersity (PDI)	Zeta Potential (mV)
PLGA-NH_2_	189 ± 1.9	0.07	−2.3 ± 0.9
Zr-PLGA-NH_2_	196 ± 4.1	0.07	−0.3 ± 0.4

## Data Availability

The data presented in this study are available in this article and Appendix A. The raw data are available upon request.

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
