# Peer review of "In Vivo PET Imaging of Monocytes Labeled with [89Zr]Zr-PLGA-NH2 Nanoparticles in Tumor and Staphylococcus aureus Infection Models"

_cancers, 2021, doi:10.3390/cancers13205069_

Round 1

Reviewer 1 Report

The manuscript submitted to Cancers by M. Krekorian et al. provides data characterizing previously described poly)lactic-co-glycolic acid with terminal amine (PLGA-NH2) nanoparticles now labelled with [89Zr]Zr as useful way to label therapeutic cells for subsequent in vivo cell tracking.  A thorough NP characterization was conducted with respect to particle stability and retention of the 89Zr along with efficiency of cell culture labeling experiments.  Lastly, an in vivo proof of principle cell tracking experiment was performed demonstrating 1000s of cells could be tracked to the site of a bacterial infection and to a tumor.   

This reviewer has no major concerns regarding this manuscript.  There are several minor concerns that could be addressed. 

  1. In the introduction,  comparisons to SPECT are made but none to other cell tracking methods the authors are familiar with. This including cellular MRI and magnetic particle imaging (MPI) for tracking SPIO NP labeled cells.  MPI, although not ready for human imaging, can also detect cell numbers in the same range of PET.  A comment by the authors in the introduction or discussion would be provide a more balanced and appropriate view of where PET sits in the field of cell tracking and sensitivity of detection.  
  2. In section 3.6 describing the [89Zr]Zr-THP-1 cell tracking by PET/MRI, the amount of [89Zr]Zr-labeled NP injected needs to be stated for comparison to the amount in the injected cells.  This information is missing and not found in the methods either. 
  3. For Figure 6, arrows or a circle around the area where the 10K cell signal is located would be helpful.  The 10K signal is barely visible in the 1 hr and 24 hr panels and not visible at all on the overlays. 
  4. In section 3.7, the first paragraph describing the S. aureus in vivo THP-1 cell tracking experiment, it is clearly stated that the following data is from the 24 hr time point.  This is not explicitly stated in the subsequent paragraph describing the MDA-MB-231 tumor THP-1 in vivo cell tracking experiment. 
  5. In Figure 8, it would be more convincing if an inset containing an enlarged area  containing the THP-1 PET signal could be shown, especially for the MDA-MB-231 experiment where it is hard to distinguish the actual PET signal from the signal in the other regions of the mouse being imaged. 
  6. The cell tracking experiments used the monocyte-like leukemia cell line THP-1 which have natural phagocytic and endocytic activity.  Yet, these cells did not label very efficiently and are not the immune cell type normally used for therapeutic purposes.  T, NK, CAR-T and CAR-NK are more normally used.  If monocytes are difficult to label, labeling T and NK cells will likely be more difficult.  Some comment on this aspect is needed and whether transfection agents might be useful but balanced against radio-toxicity.     

Author Response

Reviewer 1:

Comments and Suggestions for Authors

The manuscript submitted to Cancers by M. Krekorian et al. provides data characterizing previously described poly)lactic-co-glycolic acid with terminal amine (PLGA-NH2) nanoparticles now labelled with [89Zr]Zr as useful way to label therapeutic cells for subsequent in vivo cell tracking.  A thorough NP characterization was conducted with respect to particle stability and retention of the 89Zr along with efficiency of cell culture labeling experiments.  Lastly, an in vivo proof of principle cell tracking experiment was performed demonstrating 1000s of cells could be tracked to the site of a bacterial infection and to a tumor.   

This reviewer has no major concerns regarding this manuscript.  There are several minor concerns that could be addressed. 

  1. In the introduction, comparisons to SPECT are made but none to other cell tracking methods the authors are familiar with. This including cellular MRI and magnetic particle imaging (MPI) for tracking SPIO NP labeled cells.  MPI, although not ready for human imaging, can also detect cell numbers in the same range of PET.  A comment by the authors in the introduction or discussion would be provide a more balanced and appropriate view of where PET sits in the field of cell tracking and sensitivity of detection.  
  • We have improved the introduction with the following section: ‘’For direct cell labelling, specific cell (sub)types are isolated from patients and labelled ex vivo. For example, T-cells can be labelled with highly derivatized cross-linked iron oxide nanoparticles (NPs) and detected with magnetic resonance imaging (MRI)[10]. Although magnetic particle imaging (MPI) is highly sensitive technique (detection of ∼200 cells and a resolution of 1 mm), there are currently no clinically available scanners[11–14]. In contrast, positron emission tomography (PET) is commonly applied in clinical practice for diagnosis, staging and response monitoring in cancer and other diseases.’’ (page 2, lines 63-69)

  1. In section 3.6 describing the [89Zr]Zr-THP-1 cell tracking by PET/MRI, the amount of [89Zr]Zr-labeled NP injected needs to be stated for comparison to the amount in the injected cells.  This information is missing and not found in the methods either. 
    • The amount of [89Zr]Zr-PLGA-NH2 NPs in Matrigel was 1.56 ± 0.47 µg. This is mentioned in section 2.14.

  1. For Figure 6, arrows or a circle around the area where the 10K cell signal is located would be helpful.  The 10K signal is barely visible in the 1 hr and 24 hr panels and not visible at all on the overlays. 
    • We have added arrows in figure 6 for further clarification of the location of the signal.

  1. In section 3.7, the first paragraph describing the S. aureus in vivo THP-1 cell tracking experiment, it is clearly stated that the following data is from the 24 hr time point.  This is not explicitly stated in the subsequent paragraph describing the MDA-MB-231 tumor THP-1 in vivo cell tracking experiment. 
    • We have stated 24 hours in the text for the MDA-MB-231 tumour paragraph.

  1. In Figure 8, it would be more convincing if an inset containing an enlarged area containing the THP-1 PET signal could be shown, especially for the MDA-MB-231 experiment where it is hard to distinguish the actual PET signal from the signal in the other regions of the mouse being imaged. 
    • We added an zoom in of the signal area in the image for improved visibility.

  1. The cell tracking experiments used the monocyte-like leukemia cell line THP-1 which have natural phagocytic and endocytic activity.  Yet, these cells did not label very efficiently and are not the immune cell type normally used for therapeutic purposes.  T, NK, CAR-T and CAR-NK are more normally used.  If monocytes are difficult to label, labeling T and NK cells will likely be more difficult.  Some comment on this aspect is needed and whether transfection agents might be useful but balanced against radio-toxicity. 
    • THP-1 cells were chosen in this study as they are precursor cells for macrophages, a cell type that is highly present in inflamed areas, and are not highly differentiated with reduced receptor expression on the cell surface for pathogen recognition. Furthermore, monocytes are an intermediate phagocytic cell, between macrophages and T-cells. Although the labelling efficiency was low for the THP-1 cells, the radionuclide load was sufficient for tracking and in the range of literature. Indeed, for less phagocytic cells, labelling needs to be optimised for optimal radionuclide load. This is considered for future studies with specific peptides coating the particles. We have added this to the discussion part (page 14, lines 806-811).

Reviewer 2 Report

It is a valuable research for highly sensitive cell tracking using PET/MRI in cell-based targeted therapy. The authors developed [89Zr]Zr-PLGA-NH2 NPs for ex vivo cell labelling and in vivo cell tracking using PET/MRI. The 89Zr-labelled NPs can retain the radionuclide well over a period of 2 weeks in PBS and human serum in vitro. But immune cell THP-1 labelled with [89Zr]Zr-PLGA-NH2 NPs does not show obvious targeting performance in vivo tracking of small amount of cells in two disease models (Staphylococcus aureus infection and infiltration into MDA-MB-231 tumor). Therefore, it is necessary to further improve the targeting properties of 89Zr-labelled NPs for cell-based tracking using PET/MRI.

I would recommend reject the article to publish in this journal Cancers at present.

In addition, there are some details need to revise in the manuscript. As shown below.

1、In line 178, “2.10. Cell counting. Cell numbers after an…” There is no line break after the sentence “2.10. Cell counting.”

2、The scan bars in the In vivo PET/MRI tracking image are unclear, such as in Figure 3, 6 and8.

Author Response

Reviewer 2:

Comments and Suggestions for Authors

It is a valuable research for highly sensitive cell tracking using PET/MRI in cell-based targeted therapy. The authors developed [89Zr]Zr-PLGA-NH2 NPs for ex vivo cell labelling and in vivo cell tracking using PET/MRI. The 89Zr-labelled NPs can retain the radionuclide well over a period of 2 weeks in PBS and human serum in vitro. But immune cell THP-1 labelled with [89Zr]Zr-PLGA-NH2 NPs does not show obvious targeting performance in vivo tracking of small amount of cells in two disease models (Staphylococcus aureus infection and infiltration into MDA-MB-231 tumor). Therefore, it is necessary to further improve the targeting properties of 89Zr-labelled NPs for cell-based tracking using PET/MRI.

  • In this proof-of-concept article, we labelled THP-1 cells, a common monocyte cell line, ex vivo with [89Zr]Zr-PLGA-NH2 NPs for in vivo tracking with PET. For future studies, it would be interesting to label more specific cell types, such as T-cells. We have included this to the discussion: ‘However, future studies are needed to demonstrate feasibility of radiolabeling of other cell types such as T-cells. One strategy to enhance overall cellular uptake would be to modify the coating of NPs with eg. cell penetrating peptides or lipofectamine[42–44]. Alternatively, to improve labeling of specific subsets of immune cells, NPs can be decorated with antibodies or peptides with the desired specificity[45,46].’’ (page 14, lines 806-811)

I would recommend reject the article to publish in this journal Cancers at present.

In addition, there are some details need to revise in the manuscript. As shown below.

1. In line 178, “2.10. Cell counting. Cell numbers after an…” There is no line break after the sentence “2.10. Cell counting.”

  • We have introduced a line break after the sentence.

2. The scan bars in the In vivo PET/MRI tracking image are unclear, such as in Figure 3, 6 and8.

We have improved the scale bars in all images. Also, we improved the images for better recognition of the signals.

Reviewer 3 Report

I recommend the acceptance for publishing of the manuscript with minors corrections. "Most of the subjects related to cancer therapy are worthy of investigation.
This manuscript presents a meticulous and well-developed work.
The authors use a quite right methodology that drives to good and congruent results.
The experimental methods, results, and discussion are easy to follow.

Maybe the only thing I recommend doing is to improve the quality of the graphic material.

Author Response

Reviewer 3:

Comments and Suggestions for Authors

I recommend the acceptance for publishing of the manuscript with minors corrections. "Most of the subjects related to cancer therapy are worthy of investigation. 
This manuscript presents a meticulous and well-developed work. 
The authors use a quite right methodology that drives to good and congruent results. 
The experimental methods, results, and discussion are easy to follow. 

Maybe the only thing I recommend doing is to improve the quality of the graphic material.

  • We have improved the quality of the images.

Round 2

Reviewer 2 Report

I would recommend accept the article to publish in journal Cancers at present.